# A simple and cost-efficient automated floating chamber for continuous measurements of carbon dioxide gas flux on lakes

Kenneth Thorø Martinsen[1], Theis Kragh[1] and Kaj Sand-Jensen[1]

[1]Freshwater Biological Laboratory, Biological Institute, University of Copenhagen, Universitetsparken 4, 3rd. floor, 2100 Copenhagen, Denmark

*Correspondence to*: Kenneth Thorø Martinsen (kenneth2810@gmail.com)

**Abstract.** Freshwaters emit significant amounts of $CO_2$ on a global scale. Yet, emissions remain poorly constrained from the diverse range of aquatic systems. The drivers and regulators of $CO_2$ gas flux from standing waters require further investigation to improve knowledge on both global scale estimates and system scale carbon balances. Often, lake-atmosphere gas fluxes are estimated from empirical models of gas transfer velocity and air-water concentration gradient. Direct quantification of the gas flux circumvents the uncertainty associated with the use of empirical models from contrasting systems. Existing methods to measure $CO_2$ gas flux are often expensive (e.g. eddy-covariance) or require a high workload in order to overcome the limitations of single point-measurements using floating chambers. We added a small air pump, timer and an exterior tube to ventilate the floating chamber headspace and passively regulate excess air pressure. By automating evacuation of the chamber headspace, continuous measurements of lake $CO_2$ gas flux can be obtained with minimal effort. We present the chamber modifications and an example of operation from a small forest lake. The modified floating chamber performed well in the field and enabled continuous measurements of $CO_2$ gas flux with 40-minute intervals. Combining the direct measurements of gas flux with measurements of air and waterside $CO_2$ partial pressure also enabled calculation of gas exchange velocity. Building and using the floating chamber is straightforward. However, because an air pump is used to restart measurements by thinning the chamber headspace with atmospheric air, the duration of the air pump pause-pulse cycle is critical and should be adjusted depending on system characteristics. This may result in shorter deployment duration, but this restriction can be circumvented by providing a stronger power source. The simple design makes modifications of the chamber dimensions and technical additions for particular applications and systems easy. This should make this approach to measure gas flux flexible and appropriate in a wide range of different systems.

## 1 Introduction

Freshwaters are important components of regional and global carbon budgets (Duarte and Prairie, 2005; Raymond et al., 2013). Lakes in particular have received attention as hot spots of carbon cycling emitting $CO_2$ and $CH_4$ to the atmosphere (Tranvik et al., 2009; Holgerson and Raymond, 2016; Bastviken et al., 2011; Wik et al., 2016). Yet, the role and magnitude of carbon emissions from lakes remain uncertain as the estimated gas fluxes often depend on empirical models of gas

exchange velocity with substantial uncertainty. To apply direct measurements and improve current knowledge of drivers of temporal and spatial variability of lake $CO_2$ gas fluxes, cost-efficient and widely applicable analytical approaches are needed. Recent studies have equipped floating chambers with low cost $CO_2$ mini-loggers to quantify $CO_2$ gas flux and waterside $CO_2$ partial pressure (Bastviken et al., 2015; Natchimuthu et al., 2017). We added simple and low cost modifications to existing floating chambers, which provide automatic venting, enabling long-term and very frequent measurements of $CO_2$ gas fluxes and exchange velocities from lakes.

The diffusive flux of a gas across the lake surface (F, mmol $m^{-2}$ $h^{-1}$) can be described by the expression (MacIntyre et al., 1995):

$$F = k(C_{water} - C_{air}) \qquad \text{(Eq. 1)}$$

where k is the gas exchange velocity (m $h^{-1}$) and $C_{water}$ and $C_{air}$ is the waterside and air $CO_2$ concentration (mmol $m^{-3}$), respectively. The gas exchange velocity is influenced by near-surface turbulent mixing driven by wind shear and convection (Zappa et al., 2007; MacIntyre et al., 1995). Measurements of F and the concentration gradient ($C_{water} - C_{air}$) allow calculation of k. Empirical models of gas exchange velocity have often been parameterised from wind speed (Cole and Caraco, 1998; Crusius and Wanninkhof, 2003). This approach can potentially result in erroneous estimates of gas flux due to system scale differences in additional drivers of gas exchange velocity (Cole et al., 2010; Vachon and Prairie, 2013). For example, the contribution of convection to near-surface turbulence relative to wind shear increases with decreasing lake size (Read et al., 2012). The influence of convection on gas exchange velocity and the resulting gas flux may thus be missed if not accounted for (Holgerson et al., 2016; Podgrajsek et al., 2015). Small lakes (<0.01 $km^2$) are globally abundant and may comprise up to 20 % of the total surface area of lakes (Holgerson and Raymond, 2016) and extensive changes in $CO_2$ concentrations and vertical mixing make single or even several daytime measurements of $CO_2$ fluxes insufficient to calculate daily fluxes (Holgerson et al., 2016; Andersen et al., 2017). Increasing the temporal resolution and measuring gas flux during day and night time would enable better models of gas exchange velocity and large-scale carbon budgets.

$CO_2$ gas flux at the air-water interface can be measured from changes in $CO_2$ partial pressure over time in the headspace of floating chambers (Cole et al., 2010). Installing mini-loggers to measure air $CO_2$ concentrations has made the use of floating chambers for determination of lake $CO_2$ gas flux straightforward avoiding manual sub-sampling of chamber headspace $CO_2$ partial pressure (Bastviken et al., 2015). Chambers can be deployed for shorter time spans (15-60 min) in order to determine the gas flux, or they can be left to equilibrate with the $CO_2$ partial pressure of surface water (Natchimuthu et al., 2017). The equipment is relatively cheap and easy to use compared to other methods such as eddy-covariance (Podgrajsek et al., 2014; Jammet et al., 2017) or whole-lake addition of gas tracers (Cole and Caraco, 1998; Crusius and Wanninkhof, 2003). The disadvantage, however, is the high work load required to repeatedly lift the chambers manually to evacuate the chamber headspace before each measurement of gas flux with the floating chamber resulting in few and discontinuous measurement series (Podgrajsek et al., 2014). Some studies have developed automatic approaches to measure gas fluxes using floating chambers to increase temporal resolution and reduce the work load. Automatic systems for measurement of soil gas flux are commercially available (e.g LI-800A, LI-COR Biosciences, Lincoln, NE, USA) and rely on

automatic lifting of the chamber. For use on lakes, Duc et al. (2013) equipped a floating chamber with an inflatable balloon mounted on the chamber side to ventilate the headspace and Spafford and Risk (2018) used the forced diffusion technique which relies on passive equilibration using membranes (Risk et al., 2011). While these examples solve the mentioned problems, we wished to pursue a simpler and more cost-efficient solution to further expand the use of these methods.

To obtain high temporal resolution of $CO_2$ flux measurements from lakes, we modified the chamber described in Bastviken et al. (2015) by adding automatic venting of the chamber headspace using a small air pump, a timer and passive regulation of excess air pressure. After this improvement, the floating chamber can be left on the lake surface and provide $CO_2$ flux measurements 2-3 times every hour over several days without any manual effort. The modifications are simple yet effective and, in addition, to high frequency $CO_2$ gas flux measurements with a minimum of effort, also permit simultaneous

calculations of gas exchange velocity when $CO_2$ partial pressure in surface water and near-surface air is measured or calculated. This study adds to the growing interest and development of automatic gas flux sampling techniques by presenting a cost-efficient and simple automatic floating chamber. It was a high priority that the construction remained simple and did not require advanced technical skills and programming. We present the chamber modifications, test of performance and field data from deployment.

**2 Methods**

**2.1 Description of the chamber**

Construction, performance and use of the floating chamber with the $CO_2$ sensor ($CO_2$ ELG module, Senseair, Sweden) and battery supply (9 V) are described in detail in Bastviken et al. (2015) along with the supporting material. The chamber is simple to construct and very cheap compared to commercial alternatives (floating chamber and sensor ≈ 150-250 $ and air

20 pump and timer modifications ≈ 75-100 $). With this starting point, we added an external box containing a micro diaphragm air pump (PMDC, CTS Series, Parker, USA), a timer (VM 188, Velleman, Belgium) and battery supply (12 V, 8 x AA alkaline 1.5 V battery pack). The air pump was selected for its small dimensions (47x20x32 mm), high performance (max free flow 2.5 litres per minute) and straightforward connection. The timer allows for easy control of the air pump pulse and pause. When the air pump is on, the floating chamber is ventilated with atmospheric air through a connector in the chamber

wall through gas impermeable tubing (4 mm inner, 6 mm outer diameter). A second connector is added on the opposite chamber wall with an open, long exterior section of gas impermeable tubing (2 meter). The purpose of this outlet is to enable regulation of excess air pressure towards ambient air pressure when the air pump is on. The long tubing ensures that inward gas diffusion during measurement is negligible. Initially, we used a valve to release excess pressure when the pump was on. However, this caused build-up of excess pressure influencing $CO_2$ measurements and was abandoned. See also the

supplementary material for further information on the chamber design and parts used.

**2.2 Testing**

We tested how the air pressure within the chamber changed relative to the atmosphere when the air pump was on and ventilated the chamber. Similar to a regular field deployment, the floating chamber was placed on a water surface, but equipped with two air pressure data loggers (HOBO U20L-04, Onset Computers) placed inside and outside the chamber, which measured the absolute pressure every minute. The test consisted of two parts where the long exterior tubing was either open or closed to compare the effect on floating chamber air pressure during ventilation and measurements. This test is important because gas flux measurements may be biased if differences between the ambient and floating chamber headspace pressure occur during flux measurements with the air pump off.

In addition, passive diffusion through the long exterior gas impermeable tubing must be negligible when the air pump is off. To test this assumption, the chamber was fixed to a gas impermeable glass plate, contacts between chamber edges and glass were sealed with vacuum silicone grease and the chamber then lowered into water to make potential leakage easily detectable. This way, gas could only be exchanged through the chamber walls or the tubing. A small volume (5 ml) of pure $CO_2$ was injected through the connector on the chamber side using a syringe to increase $CO_2$ partial pressure inside the chamber (approximately six times atmospheric concentration), which was then measured for two hours with the open ended exterior tubing or with the connector closed off. We used linear regression to assess whether changes in chamber headspace $CO_2$ partial pressure occurred with time (testing the slope versus zero).

**2.3 Operation and measurements of $CO_2$ gas flux**

The floating chamber with automatic venting was deployed on a small (7260 $m^2$) forest lake in Gribskov, Denmark (lat: 55.985817 N, long: 12.271768 E) on 13 October 2017. Timer pulse and pause, air pump on and off, were 7 and 30 minutes, respectively (user defined). Atmospheric $CO_2$ partial pressure was measured 17 cm above the water surface. The $CO_2$ mini-loggers had been calibrated in $CO_2$ free air ($N_2$) following the manufacturer's guidelines. Measurements were taken every five minutes. The mixing ratio was converted to partial pressure using the daily ambient pressure recorded close by (DMI, 2017).

The flux of $CO_2$ from the lake was calculated from changes in chamber $CO_2$ partial pressure (linear slope) when the air pump was off yielding measurements of gas flux at 37-minute intervals. The $CO_2$ flux (F), reported as mmol $CO_2$ $m^{-2}$ $h^{-1}$, was calculated as (Podgrajsek et al., 2014):

$$F = \frac{dCO2}{dt} \frac{V}{RTA} \qquad \text{(Eq. 2)}$$

where the first term is the rate of change of $CO_2$ partial pressure over time in the floating chamber, V is the chamber volume (0.008 $m^3$), R is the universal gas constant ($m^3$ atm $K^{-1}$ $mol^{-1}$), T is the ambient temperature (K) and A is the chamber area in contact with water (0.075 $m^2$). Gas exchange velocity (k, m $h^{-1}$) was calculated from Eq. (1) and the normalised gas exchange velocity ($k_{600}$, m $h^{-1}$) was calculated from the ratio of Schmidt numbers (Jähne et al., 1987; Wanninkhof, 1992). Surface water $CO_2$ partial pressure was estimated from pH (pHTemp2000 MadgeTech data logger with Omega pH

electrode), water temperature (HOBO UA-002-64, Onset Computers) and alkalinity (Weyhenmeyer et al., 2012). The pH electrode was calibrated at pH 4 and 7 and subsequently corrected for drift (assumed linear). Alkalinity was measured once at deployment by acidimetric titration with 0.1 N HCl (Gran, 1952) and held constant for calculations while pH and water temperature were measured every 10 minutes. Alkalinity was 1.15 meq $L^{-1}$ and the potential bias of estimating waterside $CO_2$ partial pressure should therefore be low (Abril et al., 2015). Earlier measurements in the same system have shown very low variability in alkalinity over sub-daily time scales.

In addition to this approach, surface water $CO_2$ partial pressure was also estimated from a floating chamber with a mini-logger left to equilibrate in order to compare with the pH-alkalinity method. In this case, we used the maximum value reached after equilibration during the investigated period. Wind speed was measured 30 cm above the water surface (HOBO S-WET-A, Onset Computers) mounted on a steel peg close (<5 meters) to the floating chamber at 10-minute intervals. All analysis was performed in R (R Core Team, 2017).

## 3 Results

The addition of simple cost-efficient modifications to gas flux floating chambers allowed us to measure $CO_2$ gas flux very frequently from lake surfaces. Specifically, we added an air pump, a timer and a battery to ventilate the chamber with atmospheric air as well as a long exterior tubing to provide passive regulation of air pressure in the chamber similar to that in the ambient atmosphere (Fig. 1). The modifications ensure significant improvements over existing equipment and approach because measurement disturbance and workload are minimised and the temporal resolution is markedly increased.

When the air pump is switched on and actively ventilates the chamber (Fig. 2), internal air pressure rises compared to ambient levels (Fig. 3, a). While the increments were relative small, they may likely bias the $CO_2$ concentrations and the gas flux calculations. For this reason, we added a long exterior tubing which provided passive regulation of internal air pressure (Fig. 3, a). The length (2 meters) should ensure that excess air pressure could reach ambient levels quickly while minimising the potential $CO_2$ exchange during measurements. When testing the chamber on an impermeable surface (Fig. 3, b), no changes in chamber headspace $CO_2$ partial pressure occurred over two hours with the exterior tube on or with the connector closed off, confirming that leakage due to this modification is negligible. Performing the same test outdoors with appreciable wind exposure yielded the same result (Fig. S1). In comparison, measured rates of increase in chamber headspace partial pressure in the field were 25 to 225-fold higher.

The automated floating chamber was deployed on a small lake to test chamber operation. During a daytime period (Fig. 4) and during approximately 2.5 days (Fig. 5), $CO_2$ gas flux was measured at 37-minute intervals. At all times, the $CO_2$ flux was positive (degassing) as expected from the heterotrophic nature of humic forest lakes with $CO_2$ supersaturated surface water. The average outflux was 1.4 (range: 0.7-3.0) mmol $CO_2$ $m^{-2}$ $h^{-1}$. Measurements of waterside $CO_2$ partial pressure from the equilibration floating chamber (4250 µatm) showed relatively good agreement with the pH-alkalinity method (mean (range), 5647 (5416-5866) µatm, Fig. 4, b). We were able to obtain gas flux measurements from the system at

a temporal resolution and during periods, which, previously, were impossible with other methods or would require a high workload. Sub-daily patterns were evident in both the atmospheric $CO_2$ partial pressure and gas flux, which are likely linked to meteorological variables (Fig. 5). Changes in $CO_2$ gas flux followed patterns in gas exchange velocity (k) while the gradient in air-water $CO_2$ partial pressure was less variable. The normalised gas exchange velocity ($k_{600}$, mean (range), 0.095

(0.006-0.014) m h$^{-1}$) was significantly positively correlated with mean wind speed during the measuring interval in figure 4 (Spearman's rank, n = 19, rho = 0.64 and p < 0.01).

## 4 Discussion

We have presented a cost-efficient and easy to implement floating chamber to measure lake $CO_2$ gas flux at a high frequency. The construction of floating chambers with automatic venting mechanisms may be more or less advanced and

require different levels of technical skills (Duc et al., 2013). An advantage of the chamber presented in this study is the simple construction, low price and easy deployment. The potentially broad scale application of floating chambers could greatly improve our understanding of global scale lake gas fluxes (Tranvik et al., 2009; Raymond et al., 2013). Furthermore, the study of lake carbon balances or whole-system metabolism could be improved by including integrated measurements of $CO_2$ gas flux (Staehr et al., 2010).

Lake gas flux can be measured by different methods varying in equipment costs and required labor (Cole et al., 2010). However, the $CO_2$ mini-loggers have made measurements of $CO_2$ partial pressure in air straightforward and paved the road for non-commercial innovations, enabling scientist to improve current measurement methods (Bastviken et al., 2015). The low equipment costs would also promote deployment of several chamber units concurrently to explore spatial and temporal variations within and between sites (Natchimuthu et al., 2017). The lightweight design makes measurements

possible even in remote locations.

       Frequent and direct measurements of $CO_2$ flux are highly preferable compared to indirect methods where $CO_2$ flux is estimated as the gas exchange velocity times the $CO_2$ gradient across the air-water interface (Eq. 1). By simultaneously measuring $CO_2$ gas flux and waterside $CO_2$ partial pressure through permanently floating chambers with a small air headspace in equilibrium with surface waters, the gas transfer velocity can also be determined (Eq. 1, Fig. 4, d). The $CO_2$

gradient is usually estimated solely from $CO_2$ partial pressure in surface waters assuming a constant $CO_2$ partial pressure in the near-surface air phase similar to that in the open atmosphere. This assumption may be incorrect, particularly at low wind speeds above $CO_2$-rich ponds as shown here (Fig. 4, a) and above small sheltered streams (Sand-Jensen and Staehr, 2012). Using the same mini-loggers, $CO_2$ partial pressure can be measured just above the water surface, improving quantification of the gas transfer velocity. On the other hand, waterside $CO_2$ partial pressure may be so high under these circumstances that

the assumption of standard atmospheric partial pressures of $CO_2$ does not lead to major errors in flux calculations.

       The well-defined measurement footprint of a floating chamber makes spatial sampling possible. This may be required where spatial differences in gas transfer velocity or water $CO_2$ partial pressure are suspected (Natchimuthu et al.,

2017). A floating chamber thus provides a contrast to entire system approaches like eddy covariance methods, which measure the integrated gas flux from a larger and temporally changing measurement footprint (Jammet et al., 2017). In the numerous small lakes with a disproportionately large contribution to greenhouse gas flux from inland waters (Holgerson and Raymond, 2016), the presented floating chambers may be particularly suitable because other methods may be difficult or 5    impossible to apply (e.g. eddy covariance).

      The presented chamber showed good field performance and yielded lake $CO_2$ gas fluxes (Fig. 5, c) within the range of previously published values (Holgerson and Raymond, 2016; Torgersen and Branco, 2008; Natchimuthu et al., 2017) and similar to values found in the same system using ordinary floating chambers (not shown). The calculated gas transfer velocity was low compared to larger lakes (Holgerson et al., 2017) but similar to measures in other small lakes ($<0.01$ km$^2$) 10    using whole-lake tracer additions of propane (Holgerson et al., 2017), $^3$He and $SF_6$ (Clark et al., 1995), floating chamber connected to an IRGA (Kragh et al., 2017) and floating chambers and $CH_4$ measurements (Cole et al., 2010).

      Because the ventilation of the chamber headspace occurs through dilution with atmospheric air (Fig. 2), $CO_2$ concentrations do not always reach the ambient atmospheric values (Fig. 4 a, b). This situation can be changed by altering the duration of the air pump pulse and pause, which can be quickly modified by the user, depending on the gas flux 15    (magnitude and direction) and gas exchange velocity of the system. In our application, the large gradient in air-water $CO_2$ partial pressure meant that it was not critical that $CO_2$ partial pressure in the chamber after venting precisely reached the partial pressure in ambient air. The $CO_2$ partial pressure in the floating chambers increased linearly during flux measurements ensuring a correct rate determination not corrupted by the rising $CO_2$ partial pressure.

      Depending on the available power and study system, choosing an appropriate duration of the air pump pulse is 20    critical and should be tuned depending on the expected characteristics of the system ($CO_2$ partial pressure, $CO_2$ flux and gas exchange velocity). At the extreme end, in systems with very high gas exchange velocities the presented solution may not be suitable if $CO_2$ is released from the water to the chamber headspace almost as fast as $CO_2$ is pumped out. It is desirable to reach or approach background levels of $CO_2$ partial pressure in the atmosphere during an air pump pulse. One consequence of too short air pump pulse duration is that the elevated chamber headspace $CO_2$ partial pressure introduces a carry-over 25    effect potentially biasing repeated measurements. The solution is to increase the rate or duration of the air pump pulse. If increasing the power consumption of the air pump results in too short deployment duration an alternative stronger power source can be used. As an example of power consumption in the presented field deployment with a 30 min pause and 7 min pulse (air pump consumption of ~1.5 W or 125 ma [milli-ampere]) or approximately 39 cycles per day, the expected duration of deployment is 3.5 days with the batteries used here (~2000 mah [milli-ampere hours]). Increasing the duration of the air 30    pump pulse to 10 or 12 minutes in order to come closer to background levels of $CO_2$ partial pressure, would result in an expected deployment of 2.6 and 2.3 days, respectively. This would still permit 93 or 79 consecutive flux measurements over more than two diel cycles. As the above mentioned system characteristics vary in both time and space, the data should be quality assured and gas flux calculated from measurements in the linear range as the chamber headspace's $CO_2$ partial pressure approaches the waterside $CO_2$ partial pressure.

Further modifications to the floating chamber may be considered depending on the system and purpose of investigation. Chamber dimensions can be changed to increase the area to volume ratio, which can reduce the time required for performing a gas flux measurement and, in turn, allowing for increased temporal sampling resolution. The same can be considered for permanent chambers left on the water surface to equilibrate. The choice of dimension may be a trade-off between measurement time and longer-term stability of the floating chamber on the water surface. In this case, equilibration of the chamber headspace took several hours due to the low gas exchange velocity, and chamber dimension changes would have been necessary had the measurements of waterside $CO_2$ partial pressure relied on this method only. In this setting, the waterside $CO_2$ partial pressure calculated from pH and alkalinity likely gives a better picture of the actual levels at a given time point compared to the floating chamber where the signal is integrated over a long time period. Furthermore, the slight difference between the two methods could also be a result of spatial variability likely promoted by low rates of mixing and relative high rates of $CO_2$ production.

To contain the air pump, battery and timer, we have used a small exterior box placed on the floating chamber. The objective was to minimise the weight of the box, which allows the floating chamber to move freely and reduce the surface disturbance on natural flow regimes. In the test deployment (Figs. 4 and 5) we used a battery package containing eight 1.5 V AA batteries which is sufficient for three to four days of operation. This time frame is determined by the air-pulse duration but may also be affected by ambient temperatures. Obtaining continuous time series of $CO_2$ gas flux presents a significant improvement compared to existing floating chamber measurements which are often limited to daytime and good weather conditions. Using a larger external battery wired to the floating chamber would remove potential power limitations on the duration of the deployment without compromising the temporal resolution.

To improve current knowledge of lake $CO_2$ gas flux, continuous measurement series are required in order to examine system to global scale drivers. We have presented simple modifications to automate measurements of $CO_2$ gas flux from floating chambers on lakes based on existing methods. Using this system, we have reduced the workload required to obtain continuous measurement series considerably. A simple and cost-efficient design favours the wide application of the presented floating chamber.

**Acknowledgements**

This work was supported by grants to KSJ from COWIfonden for the study of carbon dioxide exchange on small lakes, and Carlsbergfondet for the study of small lakes. We are indebted to David Bastviken for many constructive comments to improve the manuscript during the careful review procedure.

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

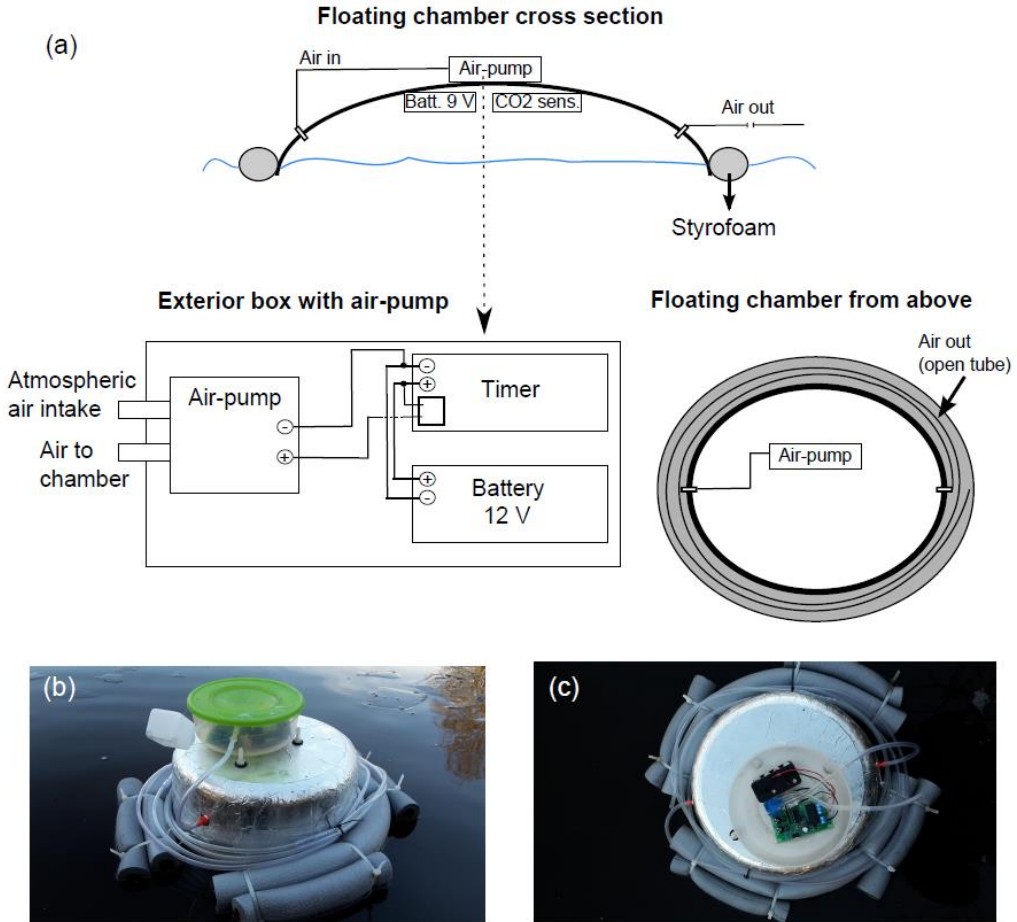

Figure 1: (a) Schematical drawing of the floating chamber with automatic ventilation showing the box, which contains air pump, timer and battery (upper part); cross-section (lower left) and view from above (lower right) of the entire floating chamber, (b) picture showing the floating chamber deployed on a lake, and (b) floating chamber viewed from above showing the exterior box containing air pump and timer.

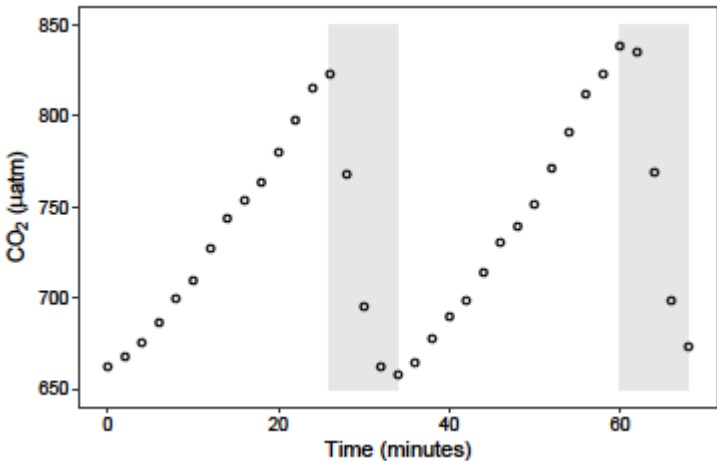

**Figure 2: Example illustrating the chamber headspace CO$_2$ partial pressure (y-axis, µatm) during two pause and pulse (grey boxes) cycles. Data from a field deployment (22 September) in the described system with a 2 minute logging interval.**

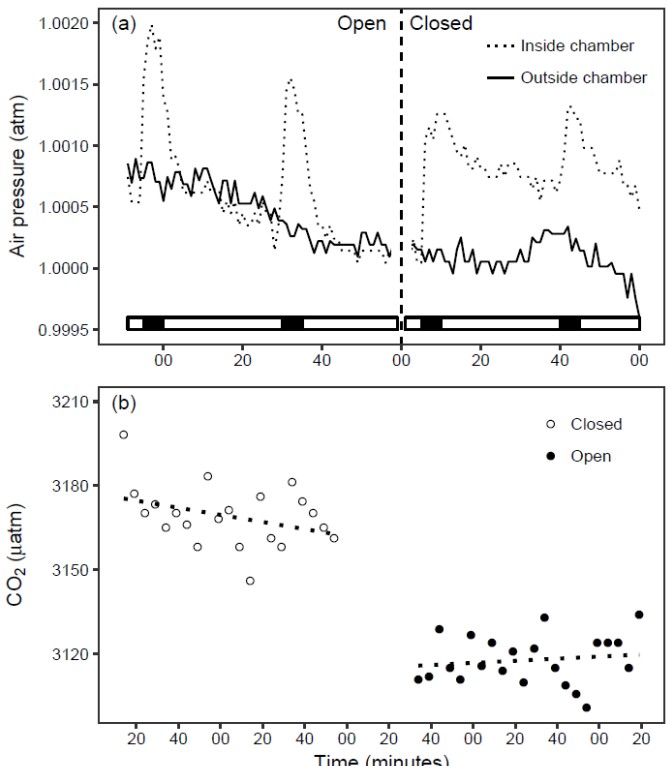

**Figure 3: (a) Air pressure measured inside (dotted line) and outside (solid line) the chamber every minute during air pulse (solid bar) and pause (open bar). Measurements are shown with and without the long (vertical dashed line) exterior tubing, which allows for passive regulation of chamber headspace air pressure. (b) Leakage test of the floating chamber showing headspace $CO_2$ partial pressure (y-axis, µatm) with the long external tubing for equilibration (open, solid points) and with the connecter closed off (closed, open points). Both regression lines are not significantly different from zero (reported as slope (±S.E, µatm $min^{-1}$), t-value, df, significance, open: 0.037 (±0.06), 0.63, 20, n.s, closed: -0.127 (±0.08), -1.66, 19, n.s).**

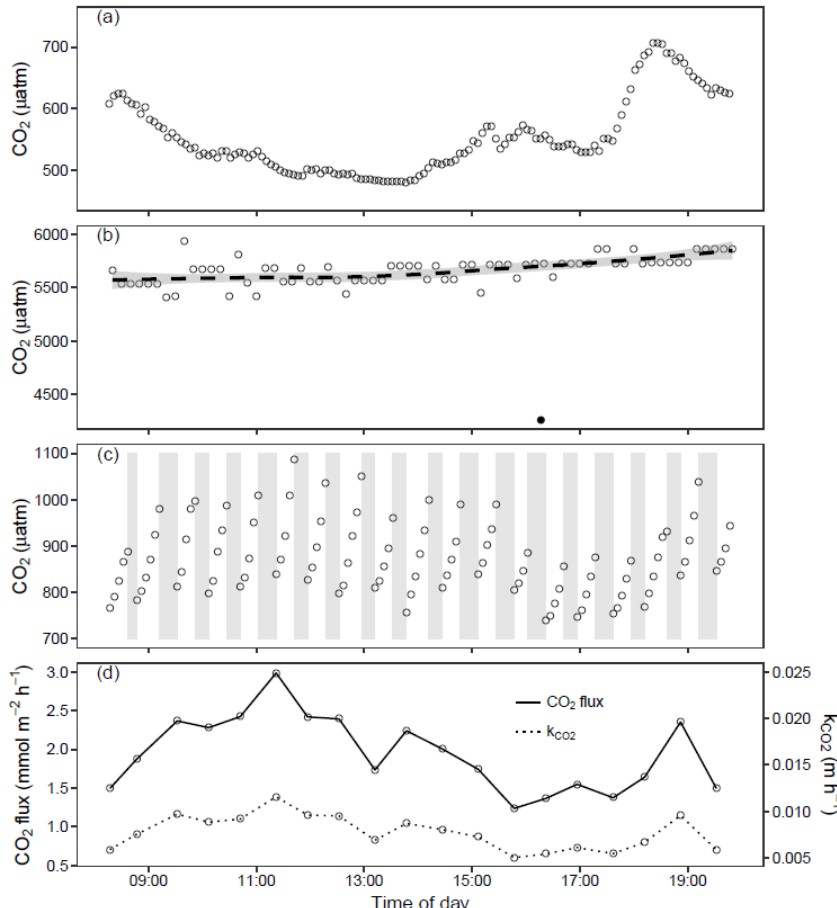

**Figure 4: Data from field deployment (14 October) showing (a) atmospheric CO$_2$ partial pressure (µatm) measured 17 cm above the water surface, (b) waterside CO$_2$ partial pressure (µatm) estimated from pH, water temperature and alkalinity (open points) fitted with a LOESS smoother and inferred from a floating chamber left to equilibrate with surface water (solid point), (c) headspace CO$_2$ partial pressure (µatm) in the automated floating chamber where the gray boxes show periods of ventilation between gas flux measurements, and (d) the calculated CO$_2$ gas flux (mmol m$^{-2}$ h$^{-1}$) and gas exchange velocity (k, m h$^{-1}$). The small abrupt changes in estimated waterside CO$_2$ partial pressure (b) are not real, but caused by minute pH changes of 0.01 unit (i.e. the resolution of pH measurements).**

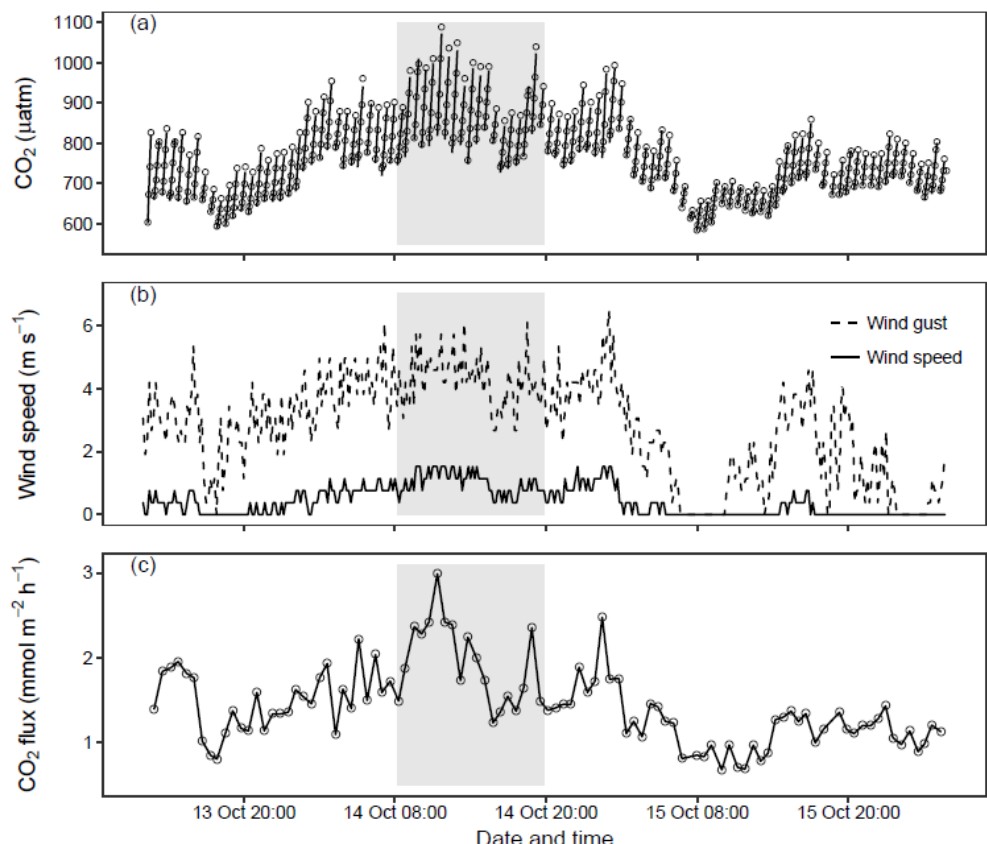

**Figure 5. Data from field deployment (13-15 October) including the time period depicted in figure 3 (gray box). (a) the headspace $CO_2$ partial pressure (µatm) in the automated floating chamber during flux measurement when the air-pump is off (open points) fitted with a linear regression (solid lines), (b) the wind speed (solid line, m s$^{-1}$) and wind gust (dashed line, m s$^{-1}$) measured 30 cm above the water surface and (c) the calculated $CO_2$ gas flux (mmol m$^{-2}$ h$^{-1}$)**

**Floating chamber cross section**

(a)

Air in

Air-pump

Batt. 9 V    CO2 sens.

Air out

Styrofoam

**Exterior box with air-pump**

Atmospheric air intake

Air-pump

Air to chamber

Timer

Battery 12 V

**Floating chamber from above**

Air out (open tube)

Air-pump

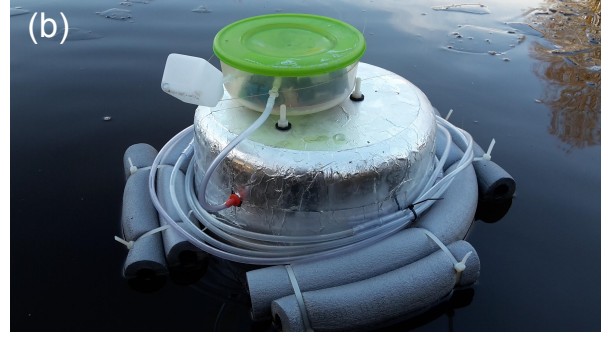

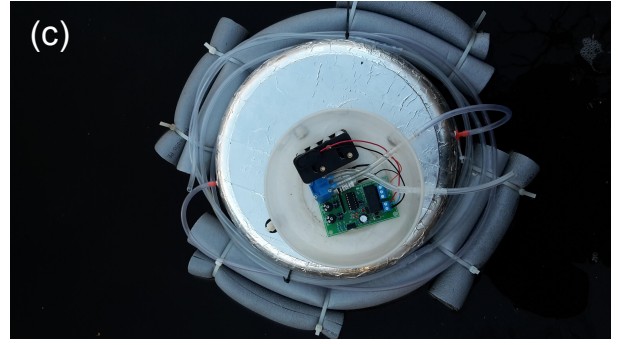

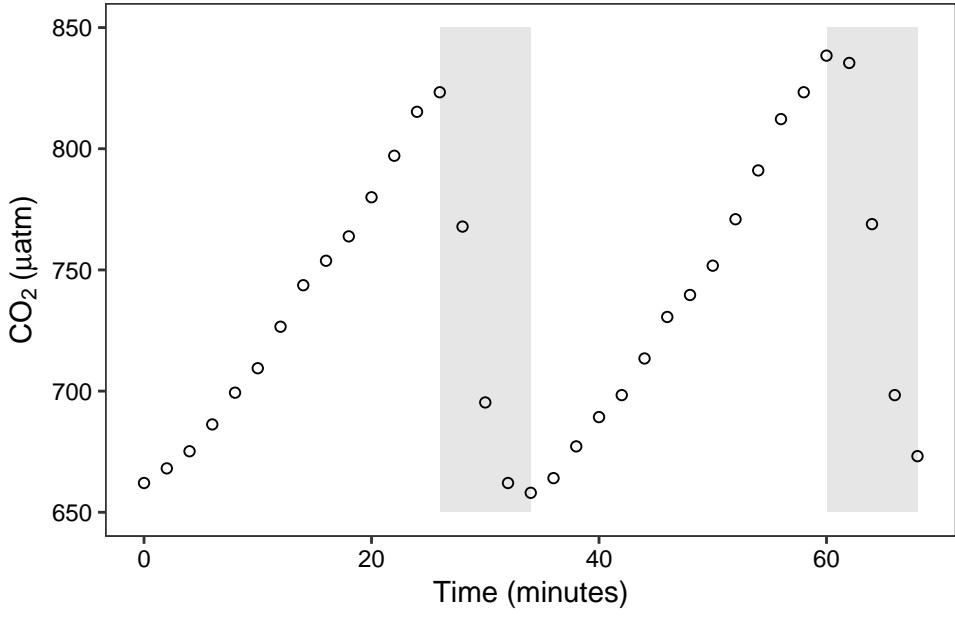

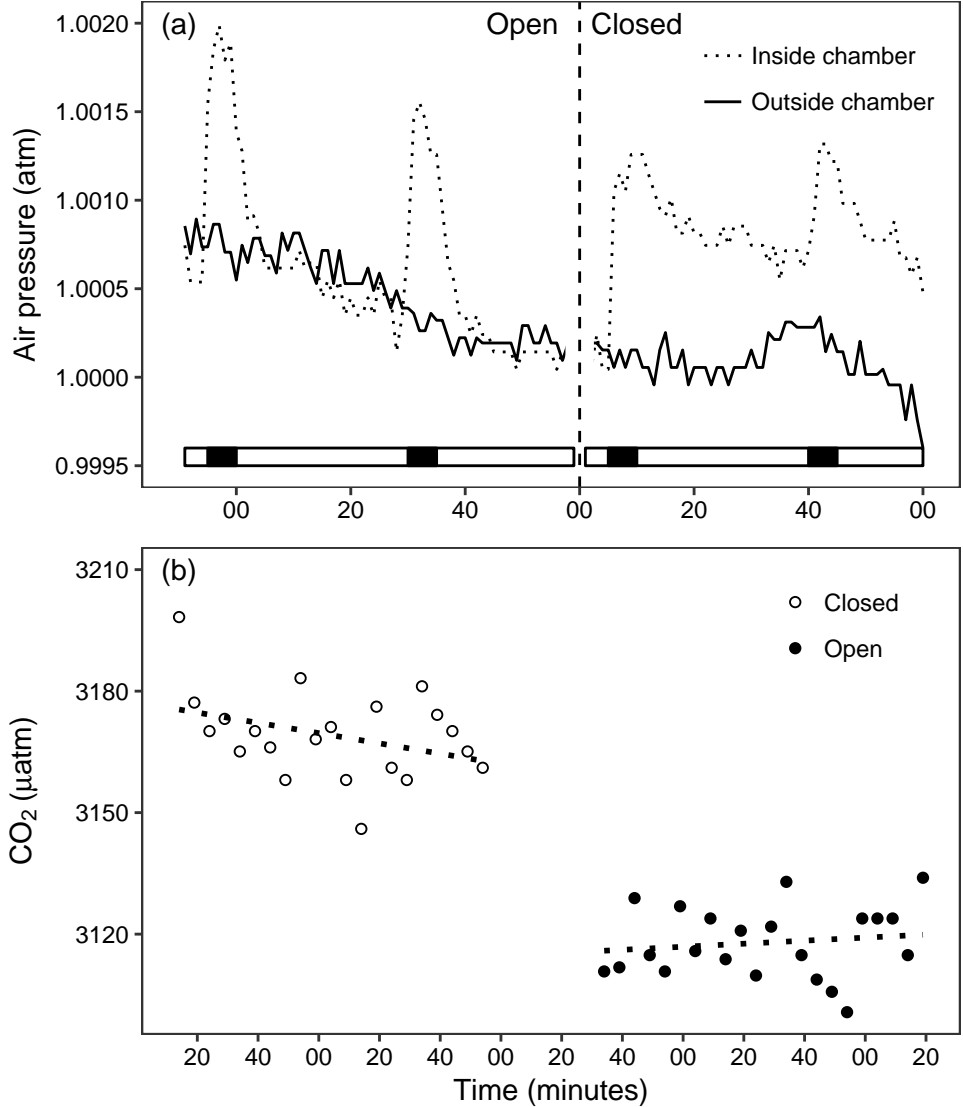

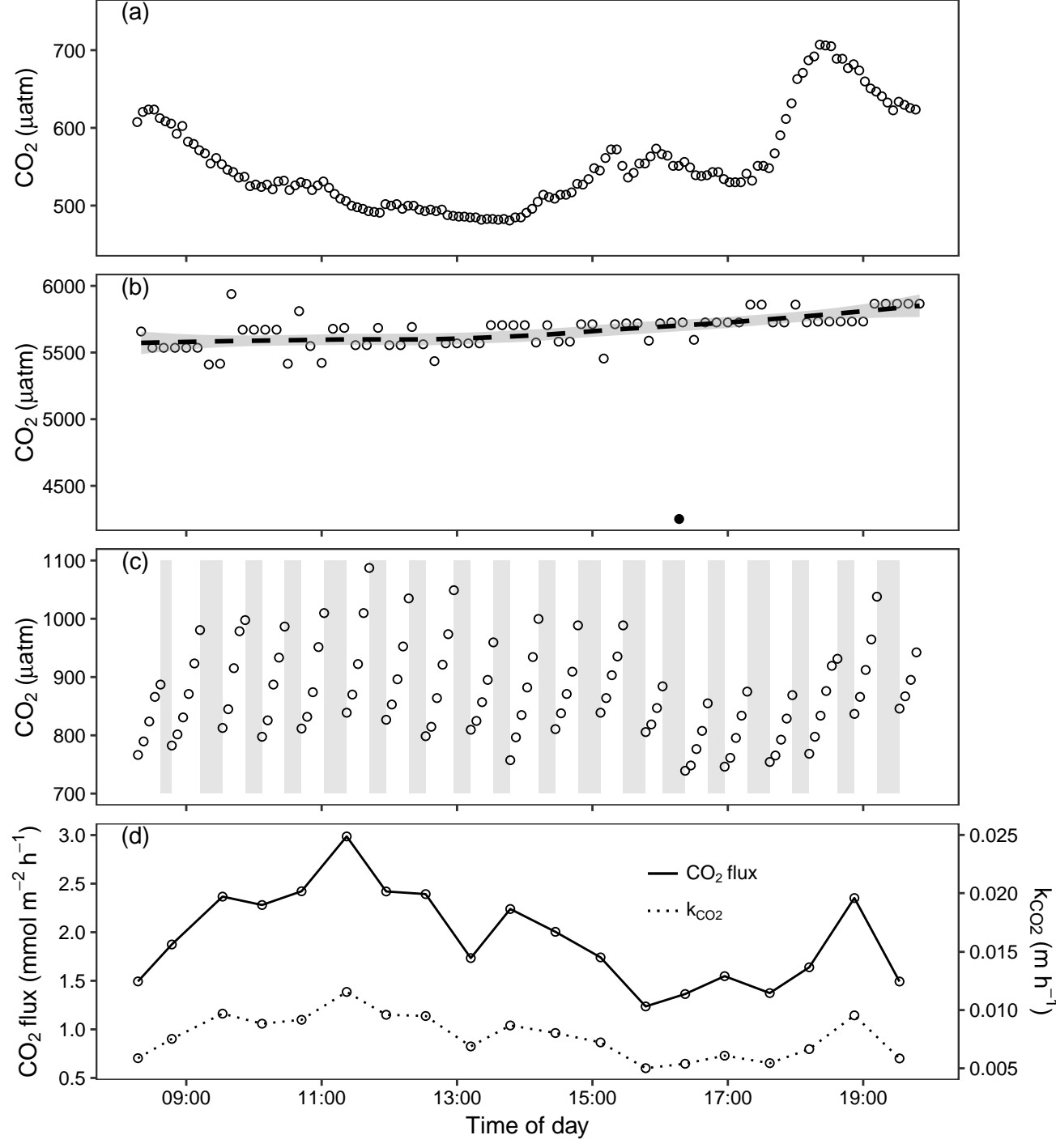

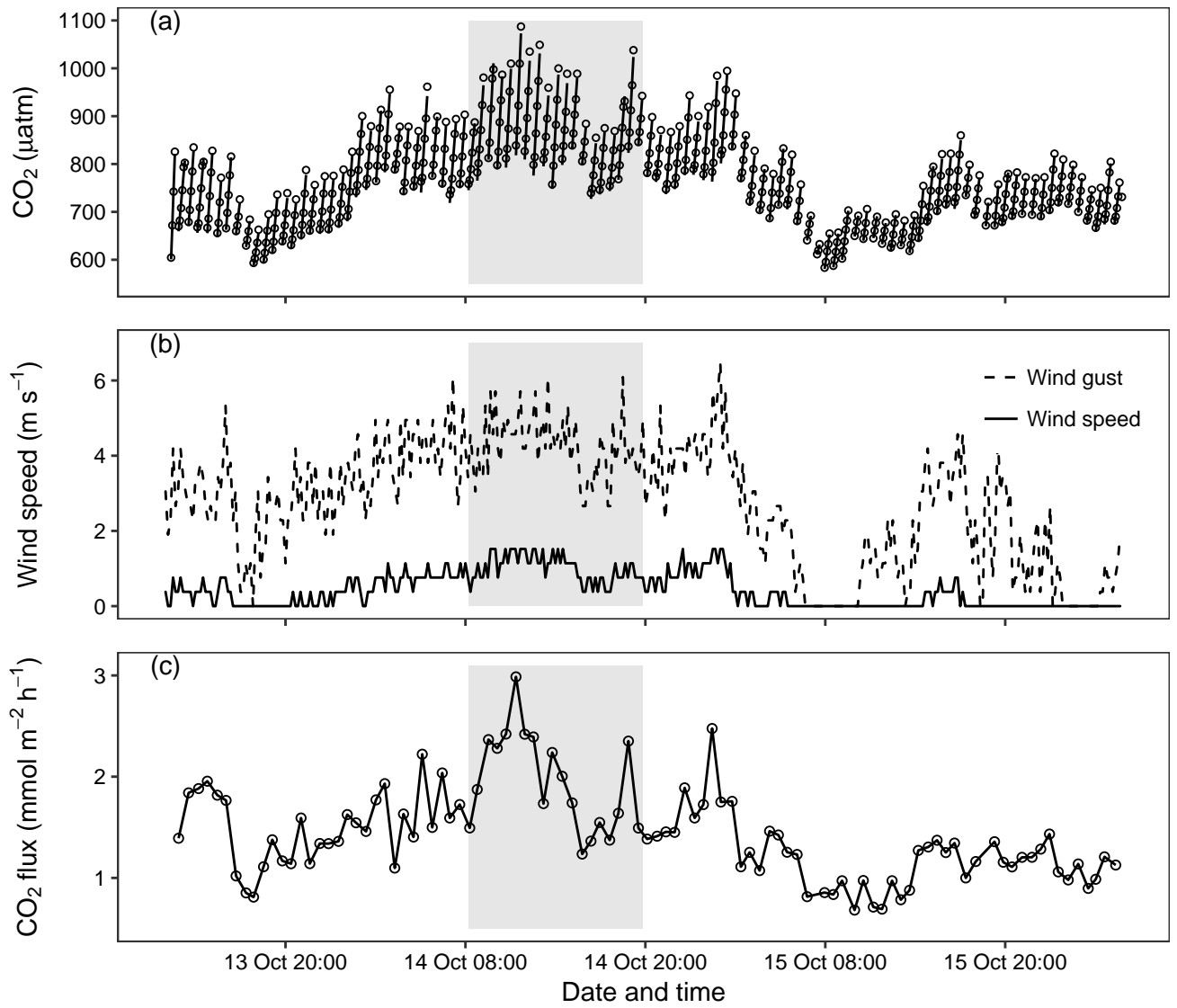