# Peer review of "A simple and cost-efficient automated floating chamber for continuous measurements of carbon dioxide gas flux on lakes"

_Biogeosciences, 2018_

## Referee Comment (RC1) · D. Bastviken (Referee) · 23 Feb 2018

This manuscript describes an approach to automatically ventilate a floating flux chamber to measure CO2 fluxes across water-air interfaces, building on a previously presented chamber with a CO2 sensor. The timer regulated ventilation of the chamber described here represents a development to restart the measurement time periods for automated repeated flux measurements over long periods. The proposed solution for chamber venting appears straightforward and has a low cost which would be advantageous if working properly. in general this type of development towards simpler and more cost-effective measurements of greenhouse gas fluxes are important for im-

proved flux assessments around the world, and also small improvements in design can lead to profound progress. I think this manuscript has potential to contribute significant such improvements if the below comments can be appripriately addressed. Hence, I would first like to thank the authors for their work and interst in improving greenhouse gas measurement methods. I think this as a very important and timely field of reserach.

General comments:

1. Please describe previous work to develop chamber wenting approaches and differences relative to the suggested approach already in the introduction. One such approach is cited later in the text (Duc et al. 2012 in EST; please note that publication on the web was 2012 but the real publication was 2013 so it should be Duc et al 2013), but there are other approaches for e.g soil/plant/wetland chamber types that could be of interest to give an overview.

2. A key is the time-frame and power consumption of the chamber venting. We tried a similar approach when working with the automatic chamber development presented in Duc et al 2013 EST. At that time we found that it took rather long time to vent the chamber completely, which in turn made the pumping consume a lot of power and also resulted in a loss of measurement time (this was a main reason why we moved on with an approach that opens the chamber to reduce venting time). It would be nice to learn more about how these problems were tested and handled in this study. Seemingly in accordance to our findings, Figure 3 indicates that background $CO_2$ levels were not reached in the 7 minutes pumping time used - the minimum chamber headspace after pumping was always between 700 and 800 ppm also during periods when the background was 500 ppm. This may have been a small problem in the test case where $pCO_2$ was very high, but could lead to biased fluxes under some conditions. How long pumping time would be needed to ensure that the background levels were rached inside the chamber? What implications would this have on the power consumption of the system?

3. With respect to the above, what was the power consumption and power limitations? How long time did the solution presented here work (please give detailed specs on what batteries were used)? Would this be a suitable technique for long-term use in the field with respect to power consumption and if so, how would this be done?

4. In the proposed design the pump and battery is placed on top of the chamber. How much does this increase the chamber mass and does this influence flux rates? The desire to minimize chamber mass is mentioned in the discussion, suggesting to put larger batteries elsewhere. Could also the pump together with the battery be placed in a separate floating box next to the chamber to remove the chamber mass issue?

5. Data from the real in-situ test is given for one day only. This data is too limited for readers to evaluate the potential of the approach. Ideally data from longer time periods covering variable weather conditions should be presented. Can such data be presented? If not, how can the system performance under variable weather conditions and system characteristics be analyzed/assessed and shown convincingly in other ways? In addition, please give real measurement data from the $CO_2$ sensor to illustrate variability in raw data (not smoothed curves as in Fig 3c).

6. Is the test of gas transport through the long open pressure equilibration tube valid for all weather conditions? Could e.g. wind-induced pumping effects or convection cause more rapid gas transfer than in the laboratory environment? How to not risk any substantial gas transfer under any weather conditions?

7. Please expand the technical description and give full details, perhaps as supplementary information. For example, please provide a detailed step-by-step guide on how to build the system with instructive pictures and component lists. A key for widespread use is easy access to such details and good instructions for persons with no background knowledge in e.g. electronics.

Specific comments:

8. Equation 2 made me a bit confused: If the term dC/dt is the change in partial pressure (atm) over time - is it then really correct to multiply with the ambient pressure (Pamb)? This would lead to atm*atm in the unit later on. If I understand this correctly, the multiplication with Pamb makes sense to me only if dC is change in molar fraction, i.e. (ppm/10ˆ6) over time.

9. Page 6, line 13-15: I am not sure I understand this sentence. Schilder et al. 2013 (Spatial heterogeneity and lake morphology affect diffusive greenhouse gas emission estimates of lakes, Geophysical Research Letters) showed that it is important to consider local k variability on lakes. It seems like the sentence is saying the opposite?

If these points can be addressed convincingly, I think this manuscript makes a good and important contribution towards one way of improving aoutomatic flux chamber design.

Best regards, David Bastviken
* * *

---

## Referee Comment (RC2) · H. Niemann (Referee) · 10 May 2018

Editior review to MS "A simple and cost-efficient automated floating chamber for continuous measurements of carbon dioxide gas flux on lakes" by Martinsen and colleagues.

Dear Kenneth Thorø Martinsen,

Unfortunately, one of the reviewers could not deliver a report because of very understandable private reasons. In order to not prolong the review process further, I will act as the second reviewer (in edition to my role as editor for this MS). The first reviewer already pointed out the most important issues, which I will not repeat here. In the following I have listed further concerns with this MS. Most importantly, the MS sometimes lacks precision (eg. the technical drawings are rather sketchy, the authors mention that the chambers are cheap but a cost estimate is only provided in the discussion). Sometimes, data are not well enough described (ie values of min/max/trend). I also miss a comparison to independent methods. The authors mention that the data are within the range of previously published data, but this seems a bit redundant in light of the large variation of $CO_2$ fluxes from individual ponds/lakes. The authors measured $CO_2$-aq/atmosphere and wind velocity, which allows calculation of fluxes (eg Wannikhof et al., 2009 and refs therein) and could be used for comparison.

The MS is generally written well and the contents fit to the scope of Biogeosciences.

Abstract P1, l9; add comma after 'often'

Intro P1, l25; add more diverse refs for lakes as ch4 and co2 source P2, l5; add ref to formula P2, l8; add more proper refs for controls on gas exchange velocity P2, l14; in comparison to what are small lakes abundant? Perhaps it's better to say: 'small lakes (XX-XX m2) are globally abundant' Better even if you could add some info as to what the total surface area of theses lakes is (ie globally) in comparison to large lakes. That would set this statement in a nice global perspective and adds to the importance of your study. P2, l29; unclear what you mean by 'pressure problems'

M&M General: improve the quality of the technical drawing of the chamber. Add all components including the $CO_2$ and T loggers (I also presume that there was an anemometer installed on top of the chamber)? There should also be references in the text to Fig. 1. In your MS, Fig. 2 is mentioned first. Also, be more precise with values you provide. E.g. why was the tubing sometimes 2 and sometimes 3m long? You often mention that the chamber is cheap. How cheap? This value comes in the discussion but is a bit out of the blue there. P3, l13. Unclear how the outlet is designed. You added a 2-3m hose connected to the chamber (I presume you used a long tubing so that leakage becomes negligible). Furthermore, you then already refer to outcomes of tests

introduced in the next section. This is a bit confusing as leak-tightness is important for the chamber design and should thus be appropriately introduced and discussed. For example, I'm missing an estimate as to how robust the measurements remain if eg small waves travel through the chamber causing a temporary volume change of the chamber's interior. This'll be equilibrated by the open tubing but the volume of the hose is limited. Thus, exchange of the chamber's interior atmosphere with the outside atmosphere may occur. P3, l29. Specify the vol of $CO_2$ that was injected. Also, how was it injected? P4, l4; lat/lon designations are incomplete (add N and E) P4, l8; provide location of the metrological station and distance to your study side. P4, l13; Is something missing in this formula? I only see the temporal change of air pressure and constants but not $CO_2$ P4, l19; elaborate how alkalinity was measured

Results General: I'm missing description of data, the reader should get a rough idea how these look like (min, max, general behaviour) - tests of $CO_2$ leakage should be shown (and not only mentioned)

Discussion General: comparison to data from other methods missing

---

## Author Comment (AC1) · 24 May 2018

We thank David Bastviken for his review. The responses to his comments, revised manuscript with track changes and new supplementary material to the manuscript are attached as supplement to this comment.

Please also note the supplement to this comment:
https://www.biogeosciences-discuss.net/bg-2018-61/bg-2018-61-AC1-supplement.zip

---

## Author Comment (AC2) · 24 May 2018

We thank Helge Niemann for his review. The responses to his comments, revised manuscript with track changes and new supplementary material to the manuscript are attached as supplement to this comment.

Please also note the supplement to this comment:
https://www.biogeosciences-discuss.net/bg-2018-61/bg-2018-61-AC2-supplement.zip

---

## Author Response (AR1)

**This document contains point-by-point responses to the comments of referee #1. Comments by**

**referee are in blue, author responses in black and changes in the manuscript in italics.**

**Referee #1, David Bastviken:**

This manuscript describes an approach to automatically ventilate a floating flux chamber to measure CO2

fluxes across water-air interfaces, building on a previously presented chamber with a CO2 sensor. The timer regulated ventilation of the chamber described here represents a development to restart the measurement time periods for automated repeated flux measurements over long periods. The proposed solution for chamber venting appears straightforward and has a low cost which would be advantageous if working properly. in general this type of development towards simpler and more cost-effective measurements of greenhouse gas fluxes are important for improved flux assessments around the world, and also small improvements in design can lead to profound progress. I think this manuscript has potential to contribute significant such improvements if the below comments can be appripriately addressed. Hence, I would first like to thank the authors for their work and interst in improving greenhouse gas measurement methods. I

think this as a very important and timely field of reserach.

We thank Dr. Bastviken for his supportive and very constructive review. We have followed his suggestions as closely as possible in the revised version.

General comments:

1. Please describe previous work to develop chamber wenting approaches and differences relative to the suggested approach already in the introduction. One such approach is cited later in the text (Duc et al. 2012

in EST; please note that publication on the web was 2012 but the real publication was 2013 so it should be

Duc et al 2013), but there are other approaches for e.g soil/plant/wetland chamber types that could be of interest to give an overview.

We agree that previous work on the subject should be mentioned earlier in the manuscript.

*We have expanded the third paragraph of the introduction. Here, we mention different ways to obtain*

*automated measurements of gas fluxes and new approaches which have been used on lakes. However, we*

*think that a more detailed overview of existing methodology used in other research (e.g soil flux studies) is*

*not within the scope of this study. The Duc et al. reference has been corrected.*

2. A key is the time-frame and power consumption of the chamber venting. We tried a similar approach when working with the automatic chamber development presented in Duc et al 2013 EST. At that time we found that it took rather long time to vent the chamber completely, which in turn made the pumping consume a lot of power and also resulted in a loss of measurement time (this was a main reason why we moved on with an approach that opens the chamber to reduce venting time). It would be nice to learn more about how these problems were tested and handled in this study. Seemingly in accordance to our findings, Figure 3 indicates that background CO2 levels were not reached in the 7 minutes pumping time used - the minimum chamber headspace after pumping was always between 700 and 800 ppm also during periods when the background was 500 ppm. This may have been a small problem in the test case where pCO2 was very high, but could lead to biased fluxes under some conditions. How long pumping time would be needed to ensure that the

background levels were rached inside the chamber? What implications would this have on the power consumption of the system?

The key to this kind of setup is indeed the power consumption of the air-pump which is the main limiting factor of the deployment duration. Using the air-pump is attractive due to the simplistic operation and installation.

The trade-off between the duration of the air-pulse and measurement period needs to be considered when deploying the chamber and is easily adjusted accordingly. We have tried longer air-pulse durations than the 7 minutes in the presented example which yields headspace concentrations closer to background levels. The pumping of air is a "thinning" process which means that headspace concentrations initially drops rapidly and then more and more slowly during an air-pulse. Ensuring background levels were reached at the start of every measurement cycle would thus require long pumping times in the order of 10-15 min. The gap between headspace concentrations after an air-pulse and ambient levels is also influenced by the flux/gas exchange velocity, from the new figure 4 (see also response to comment number 5) it can be seen that headspace concentrations are close to ambient levels during periods of low $CO_2$ flux (figure 4). Doubling the pumping time would result in approximately half the measurement time. However, we do not think that it is necessary to always reach background levels in order to determine the flux rate, especially when we use the linear increase (slope) to calculate the gas flux. If water pco2 levels are lower the response is the same as long as the headspace concentrations are changed when air is pumped through the chamber. Of course, this is only a problem when the co2 flux is from water to air.

In short, we tried to find a balance between getting close to ambient levels and a low air-pulse duration. These considerations on air-pulse duration versus deployment duration are only relevant when battery supply is limited. If a large battery on the shore, a buoy or dedicated neighboring floating chamber is supplied the duration of the air-pulse can just be increased.

*In the manuscript, we have added information on the power supply (see also comment number 3 below) and the expected deployment duration using this kind of setup. A sentence in the discussion is also expanded by mentioning what should be considered when setting the air-pump pause/pulse time. Further considerations related to these issues are also discussed in the supplementary text.*

3. With respect to the above, what was the power consumption and power limitations? How long time did the solution presented here work (please give detailed specs on what batteries were used)? Would this be a suitable technique for long-term use in the field with respect to power consumption and if so, how would this be done?

The power consumption of the air-pump is approximately 1.5 W. Using 8 standard 1.5 V AA batteries results in deployment durations of around three to four days. We definitely see this as a viable option for longer-term monitoring (> 6-7 days) but would probably use another power supply or decrease the measurement frequency.

*Information on the batteries used in the example has been added in the methods section. This information as well as additional information on the expected deployment duration with the mentioned batteries has also added to the discussion.*

4. In the proposed design the pump and battery is placed on top of the chamber. How much does this increase the chamber mass and does this influence flux rates? The desire to minimize chamber mass is mentioned in the discussion, suggesting to put larger batteries elsewhere. Could also the pump together with the battery be placed in a separate floating box next to the chamber to remove the chamber mass issue?

The increased mass on top of the chamber in the suggested design is around 400 grams. We have used traditional chambers next to the described chamber and did not find any differences in performance between the two. We have added a bit more floating material to compensate for the increased weight compared to the manually operated floating chambers. The air-pump and battery could indeed be moved away from the chamber itself. In the manuscript however we only wanted to present the simplest example of construction.

*The weights of the parts have been added in the supplementary text. Also in the supplementary text, considerations regarding placement of the air-pump/battery are discussed.*

5. Data from the real in-situ test is given for one day only. This data is too limited for readers to evaluate the potential of the approach. Ideally data from longer time periods covering variable weather conditions should be presented. Can such data be presented? If not, how can the system performance under variable weather conditions and system characteristics be analyzed/assessed and shown convincingly in other ways? In addition, please give real measurement data from the CO2 sensor to illustrate variability in raw data (not smoothed curves as in Fig 3c).

The presented data in figure 3 is during a restricted time frame in order to facilitate decent graphical presentation. However, we acknowledge that the time frame is too restricted to convincingly show the potential of this approach.

*We have, therefore, included a new figure 4, which show data from the same deployment covering a longer period as well as the time frame shown in figure 3 (marked in gray). We show raw data from the CO2 sensor (a) and calculated flux (c) along with wind speed (average and gust wind speed, b) to show the response of the floating chamber/measurements during variable wind conditions. This offers a much better impression of the performance of the floating chamber during a normal deployment in the field. It also shows the measurement response to variable mean wind speeds and wind gust).*

*The plots showing raw data from the CO2 sensor in Figure 3 (a and c) have been changed to show points instead of lines (raw data from sensor, where the molar ratio (ppm) have been converted to atmospheric partial pressure (uatm)). The data points themselves are thus unchanged, but the representation is just changed from lines to points.*

6. Is the test of gas transport through the long open pressure equilibration tube valid for all weather conditions? Could e.g. wind-induced pumping effects or convection cause more rapid gas transfer than in the laboratory environment? How to not risk any substantial gas transfer under any weather conditions i

We do not have good reason to believe otherwise. If any losses or gains due to episodic weather events occur, we should be able to see this in our measured chamber headspace CO2 concentrations as a deviation from linearity. During our field tests with deployments spread across the year (September, October, January, April and May) we have never observed that this problem show up.

7. Please expand the technical description and give full details, perhaps as supplementary information. For example, please provide a detailed step-by-step guide on how to build the system with instructive pictures and component lists. A key for widespread use is easy access to such details and good instructions for persons with no background knowledge in e.g. electronics.

We agree that detailed information was sparse in the original manuscript.

*In order to expand the technical information in a proper way, we have attached it as a supplementary text. This text gives close up pictures of the setup and notes which should facilitate easy assembly. Also in this text, is a list of parts, including weight (as part response to comment nr. 4) and suggestions on where to acquire theme.*

Specific comments:

8. Equation 2 made me a bit confused: If the term dC/dt is the change in partial pressure (atm) over time - is it then really correct to multiply with the ambient pressure (Pamb)? This would lead to atm*atm in the unit later on. If I understand this correctly, the multiplication with Pamb makes sense to me only if dC is change in molar fraction, i.e. (ppm/10ˆ6) over time.

This is indeed a mistake of ours.

*The equation has now been corrected by removing the Pamb expression.*

9. Page 6, line 13-15: I am not sure I understand this sentence. Schilder et al. 2013 (Spatial heterogeneity and lake morphology affect diffusive greenhouse gas emission estimates of lakes, Geophysical Research Letters) showed that it is important to consider local k variability on lakes. It seems like the sentence is saying the opposite?

This sentence is indeed confusing. The paragraph discusses estimation of k in relation to temporal variability of the air carbon dioxide partial pressure above the water surface. We realise that this sentence does not really add anything to this context.

*The sentence has now been removed to avoid confusion.*

**This document contains point-by-point responses to the comments of referee #2. Comments by**

**referee are in blue, author responses in black and changes in the manuscript in italics.**

**Referee #2, Helge Niemann:**

Unfortunately, one of the reviewers could not deliver a report because of very understandable private reasons. In order to not prolong the review process further, I will act as the second reviewer (in edition to my role as editor for this MS). The first reviewer already pointed out the most important issues, which I will not repeat here. In the following I have listed further concerns with this MS. Most importantly, the MS sometimes lacks precision (eg. the technical drawings are rather sketchy, the authors mention that the chambers are cheap but a cost estimate is only provided in the discussion). Sometimes, data are not well enough described (ie values of min/max/trend). I also miss a comparison to independent methods. The authors mention that the data are within the range of previously published data, but this seems a bit redundant in light of the large variation of $CO_2$ fluxes from individual ponds/lakes. The authors measured $CO_2$- aq/atmosphere and wind velocity, which allows calculation of fluxes (eg Wannikhof et al., 2009 and refs therein) and could be used for comparison.

The MS is generally written well and the contents fit to the scope of Biogeosciences.

We thank for Helge Niemann for taking the role as the second reviewer. Furthermore, we thank for the constructive comments. We have made changes in the manuscript accordingly, which have resulted in several improvements and increased the descriptive precision.

Abstract P1, l9; add comma after 'often'

*A comma has been added.*

Intro P1, l25; add more diverse refs for lakes as ch4 and co2 source

*Two references on methane emissions from freshwater and lakes added (Bastviken 2011, Science and Wik*

*2016 Nature Geoscience).*

P2, l5; add ref to formula

*A general reference for equation 1 has been added, this reference is also added for the controls on gas*

*exchange (MacIntyre 1995). We also added a reference to equation 2.*

P2, l8; add more proper refs for controls on gas exchange velocity

*Yes, see the correction above.*

P2, l14; in comparison to what are small lakes abundant? Perhaps it's better to say: 'small lakes (XX-XX m2)

are globally abundant' Better even if you could add some info as to what the total surface area of theses lakes is (ie globally) in comparison to large lakes. That would set this statement in a nice global perspective and adds to the importance of your study.

*We have added the percentage (upper limit) of the global lake surface area represented by small lakes*

*(<0.01 km2) using numbers from Holgerson and Raymond 2016.*

P2, l29; unclear what you mean by 'pressure problems'

*The mentioning of pressure problems has been deleted; it did indeed come out of nowhere, and could only*

*cause potential confusion.*

M&M General: improve the quality of the technical drawing of the chamber. Add all components including the

CO2 and T loggers (I also presume that there was an anemometer installed on top of the chamber)?

All the components are shown in the technical drawing but we agree that it is hard to recognize some of the parts. Instead of adding too many details in the overview drawing (Fig. 1), we have added a supplementary material with detailed information on the parts with accompanying pictures. The CO2 sensor also measures relative humidity and air temperature in order to correct the CO2 readings accordingly, therefore no additional temperature loggers were installed. The anemometer was installed close to the chamber on the lake, this has been clarified in the methods section now.

*Information on the placement of the anemometer during the example deployment has been added.*

There should also be references in the text to Fig. 1. In your MS, Fig. 2 is mentioned first.

We agree.

*Reference to Fig. 1 have now been added in the first paragraph of the results.*

Also, be more precise with values you provide. E.g. why was the tubing sometimes 2 and sometimes 3m long?

We have used different lengths and have not found any differences in performance. It is just important that the tubing is "long" so that diffusion is negligible.

*We have changed the value in the manuscript to 2 meter now (deleted the "3") to avoid potential confusion.*

You often mention that the chamber is cheap. How cheap? This value comes in the discussion but is a bit out of the blue there.

We agree that the price range of the floating chamber and modifications should be mentioned before the discussion.

*The part from the discussion mentioning the price range have now been moved up to the method section.*

P3, l13. Unclear how the outlet is designed. You added a 2-3m hose connected to the chamber (I presume you used a long tubing so that leakage becomes negligible). Furthermore, you then already refer to outcomes of tests introduced in the next section. This is a bit confusing as leak-tightness is important for the chamber design and should thus be appropriately introduced and discussed. For example, I'm missing an estimate as to how robust the measurements remain if eg small waves travel through the chamber causing a temporary volume change of the chamber's interior. This'll be equilibrated by the open tubing but the volume of the hose is limited. Thus, exchange of the chamber's interior atmosphere with the outside atmosphere may occur.

We agree that detailed information was sparse and a supplementary text has now been added (see also response to comment 7 by David Bastviken). We also agree that the results of the test should not be mentioned before the tests are described and we have changed the wording accordingly. We have deployed the floating chamber on several occasions with variable weather conditions (see also response to comment number 6 by David Bastviken) and we have not experienced that small waves or similar should affect the measurements. If exchange between the chamber headspace and atmosphere would occur it should be visible in the raw data. The influence of for example small waves would also be related to the chamber headspace volume. The floating chamber itself is easily replaced with this kind of design, if this phenomenon is a potential problem.

*We changed the wording of the first paragraph to avoid referring to the tests introduced late on.*

P3, l29. Specify the vol of CO2 that was injected. Also, how was it injected?

*This has been clarified in the manuscript now.*

P4, l4; lat/lon designations are incomplete (add N and E)

*Directions have now been added to the coordinate.*

P4, l8; provide location of the metrological station and distance to your study side.

The meterological data (only the ambient pressure) is from the "DMI" daily archive, Danish Meterological Institute, covering the region of Copenhagen/north-Zealand and is not from a specific meterological station. This should not be a problem in our calculations as the region is small in area, and from experience, the daily data available are representative of the study area. Furthermore, slight positive/negative deviations in the ambient pressure would only have very minor influence on the final values.

*The reference in the original manuscript was missing, so this has been added with a link to the web-site.*

P4, l13; Is something missing in this formula? I only see the temporal change of air pressure and constants but not CO2

Everything should be there (but see also response to comment from David Bastviken and the removal of the pressure term), but we see how the equation could be clarified further.

*The term dC/dt is the change in CO2 partial pressure over time. This has been changed to dCO2/dt in order to clarify.*

P4, l19; elaborate how alkalinity was measured

*Alkalinity was measured by acidimetric titration; this has been clarified in the manuscript.*

Results General: I'm missing description of data, the reader should get a rough idea how these look like (min, max, general behaviour) - tests of CO2 leakage should be shown (and not only mentioned)

We agree that the general description of data was sparse.

*We have added general description of data (flux, gas transfer velocity and CO2 partial pressure) as mean, min and max values to the results section. We have added a second plot to figure 2, so the figure now shows both the pressure (a) and tightness (b) tests.*

Discussion General: comparison to data from other methods missing

We acknowledge that the comparisons are sparse. While it is hard to compare the flux values, we can only see that they are within the expected range compared to previous studies.

*We have added comparisons to other studies investigating gas transfer velocity in small lakes using different methods. We have mentioned how the calculated gas transfer velocity is in agreement with other studies on other small lakes. Both from a study using whole-lake tracer addition (propane Holgerson 2017, helium and*

*sf6 Clark 1995) and from a conventional floating chambers connected to an IRGA (Kragh 2017) or floating*
*chambers measuring methane (Cole 2010).*

[revised manuscript text omitted]

---

## Author Response (AR2)

This document contains point-by-point responses to the comments of referee #1. Comments by the referee
are in blue, author responses in black and changes in the manuscript in italics.

**Referee #1, David Bastviken:**

**Re-Review of manuscript BG-2018-61-AC1**

I have read the revised manuscript and the responses from the authors. Most of the have been adequately
addressed. However, I two remaining requests:

1. Regarding the core of my main concern – that the suggested ventilation method takes long time and
requires substantial amounts of power to reach background levels. This is acknowledged in the response
and mentioned more briefly in the revised main text, but it is important that readers get more information
about the nature of this issue and the consequences than given in the present revised version.

This is a rather serious issue because it means that the quality of the measurements will depend on the
extent of previous gas accumulation – in turn depending on all factors affecting gas accumulation levels e.g.
pCO2 and the piston velocity during the previous accumulation period which both can change rapidly
(meaning that the effect is not constant over repeated measurements and can therefore not be corrected
for easily). Hence there will be a varying carry-over-effect making subsequent measurements partly
depending on eachother. This is a fundamental statistical problem as we prefer to work with independent
measurements – not measurements being interdependent. There will therefore be differences in
measurement quality/precision among different systems and over time in ways that are difficult to predict
or correct for. In some cases, this issue may be less important, but in other cases it will be a serious issue,
and if a very large fraction of the future data on aquatic CO2 emission would suffer from these issues, it
would be a concern.

The above issue is a large enough concern to make others abandon this venting approach previously.
Nevertheless I think it would be good if this manuscript is published to clarify this alternative as it may be
fine in some cases and stimulate to further development. In the published paper it is crucial to not only
highlight how easy and inexpensive this approach to vent chambers are, but also to thoroughly inform the
readers about the concerns and the consequences for the data produced.

We agree and acknowledge that we could elaborate further on potential disadvantages of using this
approach to ventilate a floating chamber. In the manuscript we have added a new paragraph in the
discussion where these topics are described and we further emphasize the importance of the duration of
the air pulse. The above considerations have also been added to the abstract in brief. To clearly illustrate
how the chamber headspace $CO_2$ partial pressure behaves during the air pump pause pulse cycle an
additional figure has also been added (Fig. 2).

I therefore request implementation of the below in the main text:

- Please clearly state the pumping time and power consumption needed to really ensure reaching within 5
% of background levels in the described system.

*We have included the aspects of the power consumption in the presented field deployment in the discussion.*
*We have also included how the deployment duration would be influenced by increasing the air pump rate or*

*the air pump pulse from 7 to 10 or 12 minutes which we believe would be required to approximately reach the background $CO_2$ partial pressure.*

- Please provide a figure showing the chamber headspace gas concentrations over time during the venting period is needed to facilitate reader understanding of this issue.

We acknowledge that this aspect "disappears" in the existing figures due to the long time period presented and that this is an essential element that should be illustrated.

*We have provided a new figure (Fig. 2) providing an example of the chamber headspace $CO_2$ partial pressure during two pause-pulse cycles. We have used data from a previous field deployment where the logging interval was 2 minutes in order to provide an example with higher temporal resolution.*

- Provide a clear explanation that the carry-over effect and the starting level reached after a specified pumping time will vary over time and in space depending on multiple factors, that users must be aware of varying data quality and that the best practice would be to pump long enough time the get close to the natural background levels.

We agree that these aspects are important and should be communicated clearly to the reader. A new paragraph in the discussion has been added to make readers aware of potential disadvantages and considerations necessary to avoid biased measurements.

*We have added a paragraph to the discussion which expands on this subject. We emphasize that the timer setting of the air pump is critical and that too short durations may bias measurements.*

- Please clearly in brief explain this issue and need for careful consideration of pumping time in the abstract.

We agree that these considerations, which are now a significant part of the discussion, should also be emphasized in the abstract.

*We have expanded the abstract to also include the potential disadvantage of using an air pump to evacuate the chamber headspace.*

2. On the comment regarding the potential of gas transport through the open tube during measurement periods: A test with a sealed chamber amended with CO2 was made in the lab as far as I understand showing no gas exchange over the open tube. A question was if it could not be different outdoor due to e.g. wind effects. I interpreted the response that not problems had been encountered. I think this may very well be the case and this may likely not be a big problem. However, I am also not sure gas exchange causing a continuous bias would be easily detected from the measurements, and we should be careful to test these small things before many start to use the approaches. As I think this would be easily tested by just using the same approach reported from the lab, but outdoor (and maybe even by enforcing potential wind effects using e.g. a fan) during an hour or so, I would recommend to do this test and thereby confirm our belief that this is not a problem with solid data.

That is indeed correct. The test presented in the text was performed in the laboratory and showed no exchange through the open tubing. To rule out an effect of wind in the open we have now performed the experiment outdoors using the same setup. While the average wind speed was low during the measurement period, wind gust speed reached higher levels. The observed headspace concentration showed no change during the time period confirming our previous experiments indoors. This important point has been added to the manuscript where it supplements the already presented test. We have attached a new figure in the supplementary material presenting the results.

Figure S1:

[Figure]

*In the manuscript we have mentioned this new leakage test performed outdoors following the sentences*

*describing the already existing tests performed indoors. The figure has been added as part of the*

*supplementary material.*

Provided these suggestions are implemented adequately in the main text I congratulate the authors to an inspiring paper that can become valuable for increased future data collection.

We thank David Bastviken for re-reviewing the manuscript and for the constructive comments. The changes made have further improved the manuscript and further addressed possible concerns.

[revised manuscript text omitted]